# Copula Diffusion Modelling Under Marginal Constraints

**Laura Battaglia**[1]    **Tyler Farghly**[1]    **Stefano Cortinovis**[1]    **Daniel de Vassimon Manela**[1]    **Amitis Shidani**[1]

**Christopher Williams**[1]

[1]Department of Statistics, University of Oxford, Oxford, UK

Modelling complex multivariate distributions often requires trade-offs between expressivity, tractability, and control over marginal behavior. Copula models offer a principled way to decouple marginals from dependencies, but existing approaches rely on restrictive parametric families or forgo strict marginal constraints. We show that diffusion models can be adapted to learn copula representations that preserve uniform marginals through explicit regularisation. We augment the training objective with: (i) a Monte Carlo penalty that encourages the learned score to match the desired marginal constraint over the diffusion path (ii) recent advances in online diffusion schedule optimisation. Experiments on synthetic bivariate data show that our method improves sample quality and reliably enforces marginal uniformity, supporting its effectiveness for copula estimation.

## 1  INTRODUCTION

Modelling multivariate distributions lies at the heart of probabilistic reasoning. A central challenge is balancing tractability, expressivity, and interpretability, objectives that often involve some trade-offs. Modelling the full joint distribution offers maximal flexibility but can be computationally expensive and obscure structural insights, whereas conditional factorisations enable more efficient inference at the cost of direct access to marginals.

Copulas provide an alternative parameterisation by decoupling the marginal distributions from the dependency structure [Sklar, 1959]. This separation is particularly useful when marginals are subject to external constraints, such as those informed by expert elicitation, regulatory standards, or physical laws, while the dependencies remain complex and data-driven. Such scenarios are common in finance, climate modelling, and causal inference [Zhang and Singh, 2019, Úbeda Flores et al., 2017, Evans and Didelez, 2024].

Parametric copula models often struggle to represent complex or multimodal dependencies due to restrictive assumptions. Nonparametric methods offer greater flexibility, but may lack scalability or tractable sampling [Chen and Huang, 2007]. Recent copula deep generative models [Kamthe et al., 2021] typically relax strict marginal constraints, relying instead on empirically uniform inputs. Moreover, they are notoriously difficult to train, requiring extensive hyperparameter tuning, a challenge compounded by the non-smooth boundary conditions of copulas that make the underlying densities hard to model.

In this paper, we develop a principled approach for training diffusion models to learn copula representations under explicit marginal constraints. We introduce a Monte Carlo regularisation term that encourages the learned score to match the boundary behaviour of uniform marginals throughout the diffusion process. This is done by discretising the input domain and time into bins and penalising the squared difference between the average predicted score in each bin and a reference score from the theoretical marginal. The resulting loss augments standard denoising score matching, guiding the model to approximate the correct distributional score, including its divergence at the boundaries.

We make use of recent advances in online schedule optimisation [Williams et al., 2024], which is applied during training both to condition the score model and to determine the reference time points used in the marginal penalty. The method iteratively refines a time reparametrisation by minimising Stein divergence along the diffusion path, yielding a schedule that better captures the target distribution's geometry, especially near boundaries with singular marginal behaviour. This improves stability and fidelity, with schedule-aware regularisation alone producing samples that closely match the target marginals.

The resulting models support tractable sampling and inference over marginals and joint distributions, making them well-suited for applications where preserving distributional properties is essential. This includes fields like finance, climate modelling, and causal inference, where marginals are often externally constrained while joint dependencies remain complex and data-driven.

## 2 BACKGROUND

### 2.1 DIFFUSION MODELS

Diffusion models Song et al. [2021] are a form of generative model, whose aim is to produce samples from a target distribution $\mu_{\text{data}}$, given only a finite number of samples from it. First samples $\mathbf{X}_0 \sim \mu_{\text{data}}$ are passed through the forward stochastic differential equation (SDE),

$$d\mathbf{X}_t = -\frac{1}{2}\beta_t\,\mathbf{X}_t dt + \sqrt{\beta_t}\,dB_t, \quad \mathbf{X}_0 \sim \mu_{\text{data}}, \quad (1)$$

for positive $\beta \in C^1(\mathbb{R}_+, \mathbb{R}_+)$. The time reversal $\mathbf{X}_t$ obeys

$$d\mathbf{X}_t = \frac{1}{2}\beta_t\,\mathbf{X}_t - 2\beta_t\,\nabla \log p_t(\mathbf{X}_t)dt + \sqrt{\beta_t}\,d\widetilde{B}_t, \quad (2)$$

where $\mathbf{X}_T \sim p_T \approx \mathcal{N}(0,1)$ for $T$ sufficiently large and throughout $p_t$ denotes the law of the forward samples $\mathbf{X}_t$. Therefore, the task of generating new samples from $\mu_{\text{data}}$ can be solved by simulating paths of (2). To that end, we approximate the unknown drift $\nabla \log p_t$ in (2) by minimising the *score matching loss*:

$$\ell_{\text{dsm}}(s) = \int_0^T v(t)\,\mathbb{E}\left[\|s(t, \mathbf{X}_t) - \nabla \log p_{t|0}(\mathbf{X}_t|\mathbf{X}_0)\|^2\right] \mathrm{d}t,$$

where the common choice $v(t) = 1 - e^{-\omega_t}$, with $\omega_t := \int_0^t \beta_s\,ds$, is used. Notably, the schedule $\beta$ also influences training by defining the distribution over time points at which the cost function is evaluated, specifically through sampling $\omega_t \sim \beta(dt)$. Upon temporal discretisation, the loss $\ell_{\text{dsm}}(s)$ becomes dependent on both the time spacing $\omega$ and the number of discretisation points, making it clear that discretised costs arising from different schedules are not directly comparable. Consequently, the choice of schedule plays a critical role during training, as it shapes both the weighting of time steps and the fidelity of the approximate.

### 2.2 COPULAS

By Sklar's theorem [Sklar, 1959], any joint distribution $F$ over $D$ continuous random variables $\mathbf{X} = (X^{(1)}, \dots, X^{(D)})$ with marginals $(F_{X^{(1)}}, \dots, F_{X^{(D)}})$ can be expressed in terms of a copula distribution function $C$:

$$F(\boldsymbol{x}) = C(F_{X^{(1)}}(x^{(1)}), \dots, F_{X^{(D)}}(x^{(D)})), \ \boldsymbol{x} \in \mathbb{R}^D.$$

The copula distribution is associated with a copula density, $c(\cdot)$ with uniform marginal densities given through,

$$f(\boldsymbol{x}) = c(F_{X^{(1)}}(x^{(1)}), \dots, F_{X^{(D)}}(x^{(D)})) \cdot \prod_{d=1}^D f_{X^{(d)}}(x^{(d)})$$

where $f^{(d)}$ denotes the marginal density of $X^{(d)}$. See Section 2.2 for further background on copulas.

Copula density estimation allows for simultaneous identification of the marginal densities of the variables. However, enforcing strict marginal uniformity during training is crucial. Without this, the learned marginals can drift from the true univariate distributions, and the resulting joint density loses interpretability. This is especially problematic when marginal constraints are not merely modelling assumptions but domain-informed requirements.

**Generative Copula Modelling** Several machine learning approaches have been developed to flexibly learn copula models, including those targeting specific parametric families [Wilson and Ghahramani, 2010, Ling et al., 2020]. Kamthe et al. [2021] proposed using normalising flows to fit copula distributions. However, their architecture does not enforce marginal constraints, allowing learned outputs to deviate from the copula property. Manela et al. [2024] extend this approach by introducing marginal constraints on a single variable, but generalising this to multiple marginals requires imposing undesirable restrictions on the model's dependency structure. These limitations underscore the need for more flexible and principled copula models that preserve fixed marginal constraints by design. Here, we seek to resolve this by using diffusion models.

## 3 METHODS AND EXPERIMENTS

**Problem Statement** Diffusion models require learning the score function $\nabla \log p_t$ and specifying a suitable noise schedule $\beta$, which crucially impacts training and sampling quality [Williams et al., 2024]. For non-smooth densities-like uniform distributions on $[0, 1]$, common in copula models, standard diffusion introduces smoothing artefacts. A carefully chosen schedule can mitigate this.

For copulas, the marginal on dimension $d$, denoted by $c^{(d)}(F_{X^{(d)}})$, is uniform at diffusion time $t = 0$. That is, the marginal score, interpreted in distributional form, obeys $\nabla_{x^{(d)}} \log c^{(d)}(F_{X^{(d)}}(x^{(d)})) = \infty \cdot \mathbf{1}_{(-\infty,0)}(x^{(d)}) - \infty \cdot \mathbf{1}_{(1,\infty)}(x^{(d)})$, as the score of a uniform density supported on $[0, 1]$. Let $c_t$ be the pushforward of the copula under the diffusion process after time $t$, so if $\mathbf{X}_0 \sim c$, then for diffused $\mathbf{X}_t$ we have $c_t := \text{Law}(\mathbf{X}_t)$. The diffused marginal score $s_t^{\text{ref}}(x_t^{(d)}) = \nabla_{x^{(d)}} \log c_t^{(d)}(F_{X_t^{(d)}}(x_t^{(d)}))$ is the diffusion of the score $\nabla_{x^{(d)}} \log c(F_{X^{(d)}}(x^{(d)}))$ for all $i \in \{1, \dots, D\}$.

We need to ensure that this boundary constraint for our approximate $s_t^{(d)}$ is obeyed for all $t$ in our learned model. Further, we need to ensure that our choice of schedule minimises numerical artefacts. We develop a Monte Carlo regularisation technique that relies on a well-chosen schedule to enforce this constraint. A good schedule enables the sharp boundary cutoff to be accurately learned, making the regularisation effective.

## 3.1 ONLINE SCHEDULE ESTIMATION

We adopt the approach introduced by Williams et al. [2024] to optimise the diffusion schedule while simultaneously learning the score function. Doing so allows us to mitigate tuning the scheduling parameter, and further, allows us to learn intricate densities which naturally arise in the copula regime without the need for hyper-parameter search. Given a potentially non-optimal schedule $\beta$, time is reparametrised via the dilation function $\omega_t = \int_0^t \beta_s \, ds$. Defining the dilated densities $q_{\omega_t} := p_t$, identifying an optimal schedule is equivalent to choosing an optimal time spacing $\omega$ through application of the chain rule. We will use $\omega \in C^1(\mathbb{R}_+, \mathbb{R}_+)$ for the increasing time dilation function and $\omega \in \mathbb{R}_+$ as a placeholder for its evaluates. Let $v(t')$ be from the diffusion loss, Williams et al. [2024] showed that optimal scheduling minimises:

$$D_{\text{Stein}}(q_{\omega_{t'}} || q_{\omega_t}) = v(t') \, \underset{p_t}{\mathbb{E}} \left\| \nabla \log \frac{p_{t'}}{p_t} \right\|_2^2, \quad (3)$$

across the diffusion path. Let $\{\omega_j = \omega(t_j)\}_{j=0}^{N_{\text{diff}}}$ be a discretisation of time with $\omega_{N_{\text{diff}}} = T$, the final diffusion time. The quantity

$$\lambda(\omega) := \sum_{\omega_{t_j} \leq \omega} \sqrt{D_{\text{Stein}}(q_{\omega_{t_{j-1}}} || q_{\omega_{t_j}})} \quad (4)$$

converges to the continuous path length in the fine-mesh limit and is invariant under time reparametrisation. Crucially, $\lambda$ can be estimated from any discretisation. To obtain the optimal time dilation $\omega^*$, one uses the inverse of $\lambda$:

$$\omega_t^* = \lambda^{-1}(\lambda(T) \cdot t). \quad (5)$$

Online schedule estimation proceeds by first learning the score function, estimating $\omega \to \lambda(\omega)$ using the current score, computing $t \to \omega^*(t)$, and updating the schedule using an exponential moving average with strength $\alpha \in (0,1)$. Unlike prior work which parametrises $p_t$, we parametrise $q_{\omega_t}$ so that the neural network can account for time reparametrisation directly.

**Pseudocode: Online Schedule Estimation**

```
1: for each training step do
2:     Sample x_0 ~ μ_data, t ~ U({t_0, ..., t_N}) on batch
3:     Estimate score loss ℓ_dsm(s_θ) and update s_θ
4:     if schedule update step then
5:         Estimate Stein divergences        ▷ Equation (3)
6:         Estimate ω ↦ λ(ω)                  ▷ Equation (4)
7:         Compute ω_t* = λ⁻¹(λ(T) · t)       ▷ Equation (5)
8:         ω_t ← (1 − α) ω_t + α ω_t*         ▷ α ∈ (0, 1)
9:         β_t ← d/dt ω(t)                    ▷ Finite difference
10:    end if
11: end for
```

## 3.2 MARGINAL SCORE REGULARISATION

We propose learning a copula with a diffusion model, focusing on capturing variable dependencies while ensuring uniform marginals on $[0,1]$. To assert $\mu_{\text{data}}$ has uniform marginals in our model, we regularise the learned score to posses this property in generated samples.

To achieve this, we add a regularisation term to the denoising score matching loss, encouraging the marginals $s_t^{(d)}$ to remain approximately uniform during diffusion, granting

$$\mathcal{L}(s; \omega, N_{\text{diff}}, \mathcal{B}) = \ell_{\text{dsm}}(s; \omega, N_{\text{diff}}) + \tau \cdot \ell_{\text{cop}}(s; \mathcal{B}), \quad (6)$$

where $\ell_{\text{cop}}(s; \mathcal{B})$ estimates our marginal score and compares this to a reference score function $s_t^{\text{ref}}$ with strength $\tau > 0$, on a time and spatial discretisation $\mathcal{B}$.

For our approximated score $s$, we are required to estimate our marginal score with only access to the joint score and jointly distributed samples. Using Bayes rule we can derive the marginal score expression

$$s_{\omega_t}^{(d)}(x_t^{(d)}) = \int p_t(\mathbf{x}_t^{(-d)} | x_t^{(d)}) s_{\omega_t}(\mathbf{x}_t) d\mathbf{x}_t^{(-d)}, \quad (7)$$

where $\mathbf{x}_t^{(-d)}$ is the vector $\mathbf{x}_t$ without dimension $d$, that is $\mathbf{x}_t$ with $x_t^{(d)}$ removed. We still cannot evaluate this expression as it involve drawing conditional samples. By taking local expectations, we may transform this equation into one which only requires joint samples. Let $x \in [x_{\min}, x_{\max}]$ and $\omega \in [\omega_{\min}, \omega_{\max}]$, with grids $\{x_i\}_{i=0}^{N_x}$, $\{\omega_j\}_{j=0}^{N_\omega}$. Define intervals $I_{x,i} := [x_i, x_{i+1})$, $I_{\omega,j} := [\omega_j, \omega_{j+1})$, and bins $B_{i,j} := I_{x,i} \times I_{\omega,j}$ with the total discretisation $\mathcal{B} := \{B_{i,j}\}_{i,j=1}^{N_x, N_\omega}$. Let $\bar{x}_i = \frac{x_i + x_{i+1}}{2}$, $\bar{\omega}_j = \frac{\omega_j + \omega_{j+1}}{2}$ denote the bin centres. Taking expectations over $B_{i,j}$,

$$\underset{(x_t^{(d)}, \omega_t) \in B_{i,j}}{\mathbb{E}} s_{\omega_t}^{(d)}(x_t^{(d)}) = \underset{\mathbf{x}_t \sim p_t}{\mathbb{E}} \mathbf{1}_{B_{i,j}}(x_t^{(d)}, \omega_t) s_{\omega_t}(\mathbf{x}_t),$$

which can be estimated via Monte Carlo sampling over batches, using only access to the joint score. As the bins $B_{i,j}$ are refined, the expectations $\hat{s}^{(d)}(B_{i,j}) := \mathbb{E}_{(x_t^{(d)}, \omega_t) \in B_{i,j}} s_{\omega_t}(\mathbf{x}_t)$ converge to the score evaluated at the bin centres $(\bar{x}_i, \bar{\omega}_j)$. Thus, define the regularisation

$$\ell_{\text{cop}}(s; \mathcal{B}) := \sum_{d=1}^{D} \sum_{i,j} \left( \hat{s}^{(d)}(B_{i,j}) - s_{\bar{\omega}_j}^{\text{ref}}(\bar{x}_i^{(d)}) \right)^2. \quad (8)$$

Importantly, the construction of bins requires a time and spatial discretisation. For the spatial discretisation, we know that the training data lies in the centered unit hypercube and can be gridded in this way. To ensure accurate Monte Carlo estimates, the temporal discretisation must be chosen so that the local expectations accurately capture the marginal quantities of interest along the diffusion path. To achieve this, we align the choice of bin discretisation with the diffusion schedule — choosing $\mathcal{B}$ based on the current estimate of $\omega$ — making an effective schedule critical for the accuracy of the approximation in Equation (8).

## 3.3 EXPERIMENTS

We evaluate the proposed training strategies on four synthetic bivariate datasets; Gaussian and Clayton copulas, and on transformed-versions of the more complex and multimodal MOONS and CIRCLES data-generators from `scikit-learn` [Pedregosa et al., 2011] which have been mapped to the copula rank space using empirically marginal CDFs. Further details are in Appendix B.1.

We evaluate three principal variants of score-based generative models within the DDPM framework using a variance-preserving SDE. The first model class, **FIXED**, employs standard training with the commonly used linear, quadratic, or cosine $\beta_t$ schedules. The second, **SCHED**, uses a learned diffusion schedule optimised during training (see Section 3.1). Finally, **REG** applies marginal score regularisation to improve the alignment of learned marginals, as introduced in Section 3.2. As a baseline, we also fit parametric copulas whose family selections are selected by MLE [Dissmann et al., 2013, Czado, 2019]. Training settings are discussed in Appendix B.2.

**Quantitative Results** We evaluate performance using the energy distance [Székely and Rizzo, 2013], where lower values indicate better alignment with the true distribution. Table 1 reports scores between generated and test samples in copula space. SCHED) achieves the best results on the Moons and Circles datasets under consistent training settings. As expected, the parametric copula model performs best on samples generated from the same parametric families, serving as a natural baseline in those cases. Nonetheless, SCHED remains competitive and outperforms other learned methods on the Clayton copula.

**Impact of Marginal Regularisation** Figure 1 illustrates the effect of marginal regularisation on the learned scheduler for a Gaussian fit. Regularised models produce slightly more consistent uniform marginals and tighter sample concentration within the domain. However, Table 1 shows that performance slightly degrades in multimodal settings. Although REG still outperforms FIXED schedulers on Moons and Circles, it introduces greater dispersion across modes. These findings suggest marginal regularisation offers clear benefits but must be carefully calibrated to avoid constraining the models flexibility. Future refinements may improve its applicability across more complex distributions.

## 4 CONCLUSIONS

We introduced a diffusion-based framework for learning copula models under explicit marginal constraints. By regularising the score to match the known behaviour of uniform marginals and incorporating adaptive diffusion schedule learning, our method improves the fidelity of generated

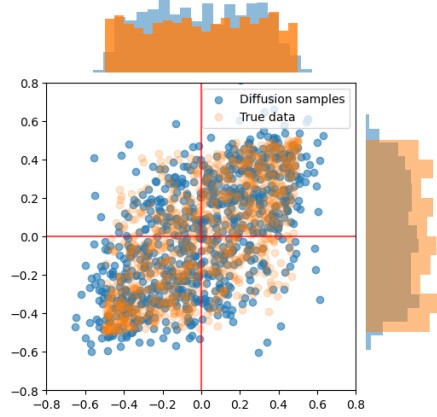

(a) Learned Schedule **without** uniform regularisation

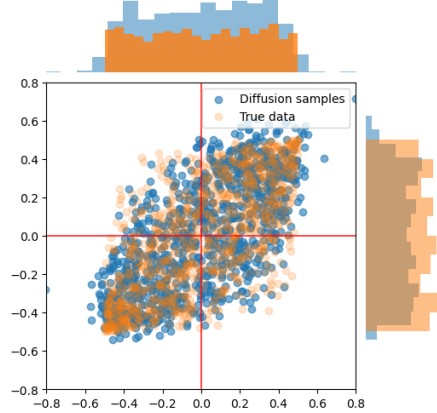

(b) Learned Schedule **with** uniform regularisation.

Figure 1: Comparison of scheduled diffusion learned with/without marginal regularisation (samples centered).

Table 1: Distances between generated and test samples in copula space across all datasets (values scaled by $10^{-3}$). The Linear, Quadratic, and Cosine methods are all part of the FIXED scheduling protocol.

| Model | Moons | Circles | Gaussian | Clayton |
|---|---|---|---|---|
| Parametric | 8.2 | 7.1 | **7.9** | **0.7** |
| SCHED | **2.1** | **1.4** | 9.7 | 1.1 |
| REG | 2.7 | 2.1 | 8.7 | 0.9 |
| Linear | 4.3 | 2.5 | 8.9 | 1.2 |
| Quadratic | 4.2 | 2.4 | 8.4 | 1.1 |
| Cosine | 2.8 | 2.6 | 10.1 | 3.7 |

samples, particularly near boundary regions where standard models often struggle. These findings suggest that principled regularisation of diffusion models offers a scalable approach to copula learning in domains requiring marginal control. Future work will extend the framework to higher dimensions and structured temporal settings.

**Acknowledgements**

LB is supported by the Clarendon scholarship. TF is supported by EPSRC EP/T5178 and by the DeepMind scholarship. SC is supported by the EPSRC CDT in Modern Statistics and Statistical Machine Learning (EP/S023151/1). DdVM is supported by a studentship from the UKs Engineering and Physical Sciences Research Councils Doctoral Training Partnership (EP/T517811/1). CW acknowledges support from a DST Fellowship and UK DTP.

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

# Copula Diffusion Modelling Under Marginal Constraints
## (Supplementary Material)

**Laura Battaglia**[1]    **Tyler Farghly**[1]    **Stefano Cortinovis**[1]    **Daniel de Vassimon Manela**[1]    **Amitis Shidani**[1]

**Christopher Williams**[1]

[1]Department of Statistics, University of Oxford, Oxford, UK

## A    COPULAS

Copulas present a powerful tool to model joint dependencies independent of the univariate margins. This aligns well with the requirements of the frugal parameterisation, where dependencies need to be varied without altering specified margins (the most critical being the specified causal effect). Understanding the constraints and limitations of copula models ensures that causal models remain accurate and consistent with the intended parameterisation.

### A.1    SKLAR'S THEOREM

Sklar's theorem [Sklar, 1959, Czado, 2019] is the fundamental foundation for copula modelling, as it provides a bridge between multivariate joint distributions and their univariate margins. It allows one to separate the marginal behaviour of each variable from their joint dependence structure, with the latter being represented by the copula itself.

**Theorem A.1.** *For a d-variate distribution function $F \in \mathcal{F}(F_{X^{(1)}}, \ldots, F_{X^{(d)}}) \in \mathbb{R}^d$, with $j^{th}$ univariate margin $F_j$, the copula associated with $F$ is a distribution function $C : [0,1]^d \to [0,1]$ with uniform margins on $(0,1)$ that satisfies*

$$F(\boldsymbol{x}) = C(F_{X^{(1)}}(x_1), \ldots, F_{X^{(d)}}(x_d)), \qquad \boldsymbol{x} \in \mathbb{R}^d.$$

*1. If $F$ is a continuous d-variate distribution function with univariate margins $F_1, \ldots, F_d$ and rank functions $F_1^{-1}, \ldots, F_d^{-1}$ then*

$$C(\boldsymbol{u}) = F(F_{X^{(1)}}^{-1}(u_1), \ldots, F_{X^{(1)}}^{-1}(u_d)), \qquad \boldsymbol{u} \in [0,1]^d.$$

*2. If $F$ is a d-variate distribution function of discrete random variables (more generally, partly continuous and partly discrete), then the copula is unique only on the set*

$$Range(F_1) \times \cdots \times Range(F_d).$$

*The copula distribution is associated with its density $c(\cdot)$*

$$f(\boldsymbol{x}) = c(F_{X^{(1)}}(x_1), \ldots, F_{X^{(d)}}(x_d)) \cdot f^{(1)}(x_1) \ldots f^{(d)}(x_d)$$

*where $f_i(\cdot)$ is the univariate density function of the $i^{th}$ variable.*

Note that Sklar's theorem explicitly refers to the **univariate marginals** of the variable set $\{X_1, \ldots, X_d\}$ to convert between the joint of univariate margins $C(\boldsymbol{u})$ and the original distribution $F(\boldsymbol{x})$. For absolutely continuous random variables, the copula function $C$ is unique. This uniqueness no longer holds for discrete variables, but this does not severely limit the applicability of copulas to simulating from discrete distributions. The non-uniqueness does play a more problematic role in copula inference, however [Genest and Nelehová, 2007].

An equivalent definition (from an analytical purview) is $C : [0,1]^d \to [0,1]$ is a $d$-dimensional copula if it has the following properties:

1. $C(u_1, \ldots, 0, \ldots, u_d) = 0$;
2. $C(1, \ldots, 1, u_i, 1, \ldots, 1) = u_i$; and further,
3. $C$ is $d$-non-decreasing.

**Definition A.1.** *A copula $C$ is $d$-non-decreasing if, for any hyperrectangle $H = \prod_{i=1}^{d} [u_i, y_i] \subseteq [0,1]^d$, the C-volume of $H$ is non-negative.*

$$\int_H C(\boldsymbol{u}) \, d\boldsymbol{u} \geq 0.$$

# B  EXPERIMENTAL DETAILS

## B.1  DATASETS AND PARAMETERIZATIONS

**Experimental setup.**    We systematically evaluate our diffusion-based copula modeling approaches on a suite of synthetic and semi-synthetic datasets, each designed to probe different aspects of dependence structure and marginal complexity.

- **Clayton Copula**: We sample points from a bivariate Clayton copula [Joe, 2014] using the `pyvinecopulib` library [Thomas Nagler and Thibault Vatter, 2021]. This is to test the model performance on an Archimedean copula with non-linear and non-Gaussian dependence.
- **Moons**: We construct a challenging non-Gaussian dataset by applying the empirical copula transformation to data generated by `sklearn.datasets.make_moons`. A large auxiliary sample ($N = 10^6$) is used to fit empirical marginal CDFs, and a smaller subset ($n = 1000$) is mapped to copula space for training and evaluation.
- **Circles**: Similarly to the Moons example, we use `sklearn.datasets.make_circles` to generate a ring-shaped distribution, mapping samples into copula space via empirical marginal CDFs estimated from a large auxiliary sample.

For each dataset, we train and compare three diffusion models: a standard Denoising Diffusion Model (DDM), a DDM with soft marginal constraints, and a DDM with reweighting to uniform marginals. All models are trained for 2,000 epochs (unless otherwise specified), with 2,000 samples drawn from each trained model for quantitative and qualitative evaluation. Hyperparameters such as the number of diffusion steps, marginal constraint strength, and model architecture are kept fixed across datasets for comparability.

## B.2  DIFFUSION TRAINING HYPERPARAMETERS

The experiments were all conducted on a MacBook M1 Pro (2023). For all experiments, we standardized the dataset size to $n = 1000$ samples for each method and ensured that the number of training epochs was consistent across all models. The following summarizes the key hyperparameters used for the diffusion schedulers and data generation processes:

- **Score Model:** All models used a neural network with $d_{\text{model}} = 20$ hidden units, 10 layers, and input dimension $x_{\text{dim}} = 2$.
- **Scheduler:** For all methods, we used $n_{\text{steps}} = 100$, minval = 1e-5, and maxval = 5 for the diffusion process.
- **Soft Constraint Penalty:** For the soft-constraint DDPM **Reg**, we set the penalty parameter $\lambda_{\text{penalty}} = 0.01$.
- **1D Score Learning:** The 1D uniform score was learned using $N_{x,\text{bins}} = 20$, $x_{\text{min}} = -3.0$, $x_{\text{max}} = 3.0$, $N_{t,\text{bins}} = 20$, $t_0 = 0.0$, and $t_{\text{end}} = 1.0$.
- **Batch Size:** Training was performed with a batch size of 512.
- **Epochs:** The number of training epochs was standardized across all methods (e.g., 2000 epochs for 2D models, 600 epochs for 1D models).