# OpenReview forum: "Copula Diffusion Modelling Under Marginal Constraints"
_auai.org/UAI/2025/Workshop/TPM — TPM 2025_

### Official Review · Reviewer_uVxu · 2025-06-15
**Improving diffusion models with marginal scores**

**Rating:** 2

**Review:**

This paper aims to improve the performance of continuous diffusion models. Specifically, in addition to learning the score alone, the paper proposes to also learn the "marginal score" (the score corresponding to univariate marginal distributions). This allows the model to regularize the score by ensuring that it matches the marginal score (the conditional expectation of the score matches the corresponding marginal score). Specifically, this regularization is done by an auxiliary loss term that minimizes the squared distance between (i) the directly-estimated marginal score, and (ii) the marginal score computed by the expected (joint) score.

While the general idea is very interesting, my main concern is whether the expected (joint) score can be estimated accurately using Monte Carlo sampling (as described in Section 3.2). For example, it seems that in Figure 1 the marginal score is not enforced very well.

I'm curious how the learned schedule in Section 3.1 and the marginal score idea relate. Additional experiments that analyze this would be very helpful.

In the second paragraph of Section 3, it is quite confusing to me what does $\mathbf{1}(-\infty, 0) (x^{(i)})$ mean.

---

### Official Review · Reviewer_bdEX · 2025-06-15
**Accept**

**Rating:** 3

**Review:**

This paper presents an approach to copula modeling based on diffusion models. A key challenge in copula modeling is enforcing uniform marginal distributions, and they propose a Monte Carlo loss term paired with (known) scheduling techniques to address this challenge. Quantitative results on synthetic bivariate data provide some justification for their approach.

**This paper is appropriate for the workshop, containing work that is relevant and interesting to the TPM community.**

As someone not too familiar with continuous diffusion, I wonder why the seemingly main issue of the sharp boundary behavior of the uniform marginal densities cannot be circumvented, say, through some data transform.

Comparison with other neural copula approaches would improve the paper (especially experiments but also just some discussion). Also, how is the method expected to scale beyond bivariate distributions?